# Topologically crafted spatiotemporal vortices in acoustics

Hongliang Zhang[1,6], Yeyang Sun[1,6], Junyi Huang[1,6], Bingjun Wu[1], Zhaoju Yang [1] ✉, Konstantin Y. Bliokh[2,3,4] & Zhichao Ruan [1,5] ✉

Vortices in fluids and gases have piqued the human interest for centuries. Development of classical-wave physics and quantum mechanics highlighted wave vortices characterized by phase singularities and topological charges. In particular, vortex beams have found numerous applications in modern optics and other areas. Recently, optical *spatiotemporal* vortex states exhibiting the phase singularity both in space and time have been described. Here, we report the topologically robust generation of *acoustic* spatiotemporal vortex pulses. We utilize an acoustic meta-grating with broken mirror symmetry which exhibits a topological phase transition with a pair of phase singularities with opposite topological charges emerging in the momentum-frequency domain. We show that these vortices are topologically robust against structural perturbations of the meta-grating and can be employed for the generation of spatiotemporal vortex pulses. Our work paves the way for studies and applications of spatiotemporal structured waves in acoustics and other wave systems.

Wave vortices, i.e., structures with the wavefield intensity vanishing in the center and the phase winding around, are of enormous importance for various areas of physics. They are essential parts of almost any structured waves: atomic orbitals and superfluids in quantum mechanics, complex wave interference from ocean waves to nanophotonics and metamaterials, etc. Cylindrical vortex beams have been generated and found applications in electromagnetic[1–8], sound[9–17], elastic[18], electron[19–21], neutron[22], and atom[23] waves. Such states contain on-axis vortex lines and carry intrinsic orbital angular momentum (OAM) along their propagation direction.

Recently, there was a great rise of interest in *spatiotemporal vortex pulses* (STVPs), which are generalizations of usual 'spatial' vortex states to the space-time domain and the OAM tilted with respect to the propagation direction[24–39]. This conforms with the rapidly growing field of space-time structured waves allowing manipulation in both spatial and temporal degrees of freedom[40,41]. In the simplest case,

STVPs are flying doughnut-shaped pulses with the vox line and OAM orthogonal to their propagation direction. Until now, STVPs have been generated only in optics, although theoretically these have also been discussed for quantum matter and acoustic waves[33]. In addition, spatiotemporal toroidal pulses with topological vortex and skyrmionic structures have also been explored recently[26,42–45].

Here, we report the topologically robust generation of acoustic STVPs for *sound* waves in air. Our STVP generator is based on a meta-grating with broken mirror symmetry, which is controlled by a synthetic asymmetry parameter[32,46]. We show that such meta-grating exhibits vortices in the transmission spectrum function in the momentum−frequency domain, which appear in pairs at the critical value of the asymmetry parameter. Transmission of a Gaussian pulse with the central parameters corresponding to the transmission-spectrum vortex imprints this vortex in the space-time domain of the transmitted pulse. Our method uses the zeroth-order transmitted

[1]School of Physics, Zhejiang Province Key Laboratory of Quantum Technology and Device, and State Key Laboratory for Extreme Photonics and Instrumentation, Zhejiang University, Hangzhou 310027, China. [2]Theoretical Quantum Physics Laboratory, Cluster for Pioneering Research, RIKEN, Wako-shi, Saitama 351-0198, Japan. [3]Centre of Excellence ENSEMBLE3 Sp. z o.o., 01-919 Warsaw, Poland. [4]Donostia International Physics Center (DIPC), Donostia-San Sebastián 20018, Spain. [5]College of Optical Science and Engineering, Zhejiang University, Hangzhou 310027, China. [6]These authors contributed equally: Hongliang Zhang, Yeyang Sun, Junyi Huang. ✉e-mail: zhaojuyang@zju.edu.cn; zhichao@zju.edu.cn

field and is independent of the vector (polarization) properties of the field; it can be applied to longitudinal sound, transverse optical, or other types of waves. Importantly, akin to topological features of electronic and optical systems[47–50], this method exploits a nodal phase-singularity line in the transmission spectrum function and hence is *topologically protected* against structural disorder of the meta-grating. Our results open the avenue for spatiotemporal vortex generation and applications in acoustics and other areas of wave physics[7,11,32,51–55].

## Results

### Breaking spatial mirror symmetry for the generation of STVPs

The idea of our STVP generator, schematically shown in Fig. 1a, is based on the spatial mirror symmetry breaking[32]. A $z$-propagating Gaussian pulse impinges on the structure (meta-grating) lying in the $z = 0$ plane and homogeneous along the $y$-axis. If the meta-grating is mirror-symmetric about the $x = 0$ plane, the phase distribution of the transmitted pulse must also be symmetric about this plane, and thus can bear no phase singularity (vortex) on the $z$-axis. Therefore, the necessary condition for generating $z$-propagating STVPs carrying a phase vortex is the mirror symmetry breaking.

We design a meta-grating shown in Fig. 1a, b (for details see Supplementary Note 1) with a unit cell consisting of four air blocks of different sizes (white areas in Fig. 1a, b). All cells are connected with a middle air channel (yellow areas in Fig. 1a, b). Starting with the four air blocks mirror-symmetric with respect to the $x = 0$ plane (Fig. 1b), we break the mirror symmetry by shifting the blocks along the $x$-axis by

$\delta x_i = A_i\eta$, where $i = 1,2,3,4$ is the block number, $A_i$ is the shifting coefficient given in Supplementary Table S1 and $\eta$ is the synthetic dimensionless parameter which characterizes the degree of the mirror asymmetry of the grating.

This asymmetric modulation of the incident pulse can be illustrated by the transmission spectrum function $T(k_x,\omega)$, where $\omega$ is the angular frequency of a plane wave, and $k_x$ is the wavevector component along the meta-grating. Here only the zeroth-order diffraction is considered in the operating-frequency range. For the mirror-symmetric case, $\eta = 0$, the complex transmission spectrum $T(k_x,\omega)$ is symmetric with respect to $k_x = 0$ (Fig. 1c). Breaking the mirror symmetry with $\eta = 0.5$, we find that the transmission spectrum function $T(k_x,\omega)$ acquires two phase singularities (vortices) with zero transmission amplitudes $|T| = 0$ in the center and opposite phase winding numbers (topological charges) $l = +1$ (white arrow) and $l = -1$ (black arrow) (Fig. 1d). For higher value $\eta = 0.7$ the two vortices are further separated from each other (Fig. 1e).

The vortices in the transmission spectrum function are topological objects in the $(k_x,\omega)$ domain, and they are always created or annihilated in pairs of opposite topological charges $l$ upon perturbations in the system. Figure 1f shows the evolution of the vortices in the $T(k_x,\omega)$ function with the parameter $\eta$. One can see that two vortices in the $(k_x,\omega)$ planes form a single nodal line $|T| = 0$ in the extended 3D space $(k_x,\omega,\eta)$. The $(k_x,\omega)$ planes with a fixed value of $\eta$ have either zero or two intersections with the nodal line, which correspond to zero or two vortices with opposite $l$. Thus, the total topological charge is a conserved quantity. We numerically find that in our meta-grating the pair

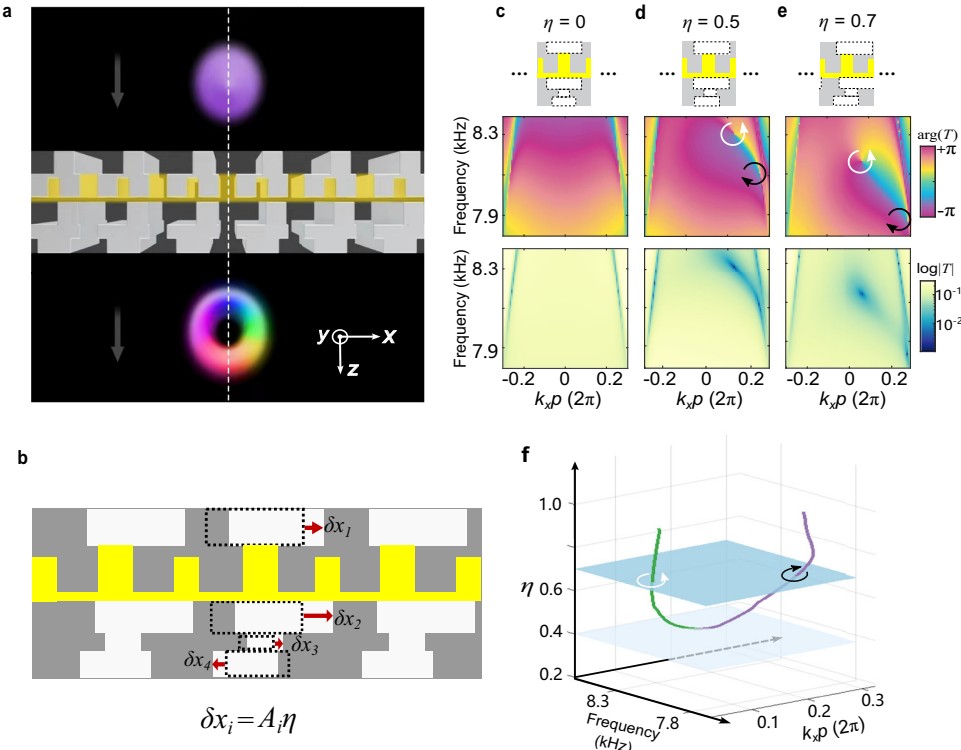

**Fig. 1 | Acoustic meta-grating for the generation of acoustic spatiotemporal vortex pulses. a** Schematics of the meta-grating generating a spatiotemporal vortex pulse for airborne sound by breaking the mirror symmetry. The period of the meta-grating is $p = 33.34$ mm. **b** The structure of the meta-grating, where the solid material is shown in gray, the position of the yellow block (air) is fixed, while the four white blocks (air) marked can be $x$-shifted to the left or right. The displacements of these blocks, $\delta x_i$, break the mirror symmetry with respect to the $x = 0$ plane, and this asymmetry is quantified by the dimensionless parameter $\eta$ (see explanations in the text). **c**–**e** Numerical simulations for the phase (top) and the

amplitude (bottom) of the transmission spectrum function $T(k_x,\omega)$ for different values of the asymmetry parameter $\eta$. Phase singularities (vortices) with the winding numbers (topological charges) +1 and −1 are indicated by the white and black arrows, respectively. **f** The pair of vortices in the panels (**d**) and (**e**) corresponds to a single phase-singularity (nodal) line in the 3D space $(k_x,\omega,\eta)$ extended by the asymmetry parameter $\eta$. The vortex pair emerges at the critical values of $\eta_c = 0.40$. This nodal line and the separated vortices are topologically protected against small perturbations in the system.

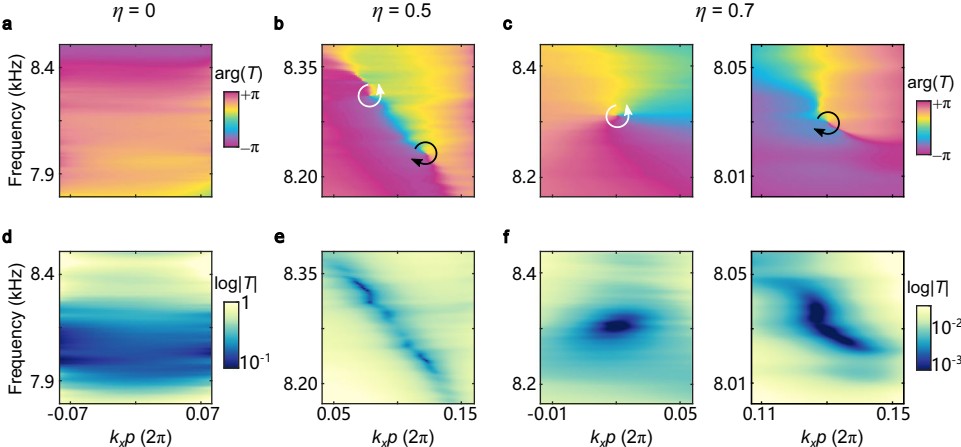

**Fig. 2 | Transmission spectrum function for different values of the asymmetry parameter $\eta$.** Experimentally measured phase (**a**–**c**) and amplitude (**d**–**f**) of transmission spectrum function for different values of the asymmetry parameter $\eta$. Vortices with the winding numbers of +1 and −1 are indicated by white and black arrows, respectively.

of vortices emerges at the critical value $\eta_c = 0.40$, where the $\eta = \eta_c$ plane touches the nodal line of $T(k_x, \omega, \eta)$.

To experimentally demonstrate this topological phase transition (i.e., the birth of the vortex-antivortex pair in the transmission spectrum), we fabricate three meta-gratings corresponding to $\eta = 0$, 0.5, 0.7 (Supplementary Figure S4) and measure their transmission spectra $T(k_x, \omega)$ ("Methods"). Figure 2 shows the measured distributions of the phase and amplitude of these transmission spectra. For the mirror-symmetric case (Fig. 2a, d), there are no phase singularities in the measured transmission spectrum. By breaking the mirror symmetry with $\eta = 0.5$ (Fig. 2b, e), two vortices with opposite winding numbers, indicated by the white and black arrows, appear at $\omega/2\pi = 8.31$ kHz and $\omega/2\pi = 8.23$ kHz. For $\eta = 0.7$ (Fig. 2c, f), the two vortices are further separated and located at $\omega/2\pi = 8.31$ kHz and $\omega/2\pi = 8.04$ kHz. These results agree that the topological phase transition at the critical value $\eta_c = 0.40$. The appearance of the vortices confirms that breaking the mirror symmetry of the meta-grating produces transmission vortices in the $(k_x, \omega)$ domain which can provide asymmetric spatiotemporal modulation of the incident pulse.

**Topologically protected generation of STVPs**

Owing to the asymmetric modulation and the Fourier-transform properties, a phase singularity of the transmission spectrum function, located at some point $(k_{0x}, \omega_0)$ in the $(k_x, \omega)$ domain, can be directly transferred into the spacetime $(x, t)$ domain for the transmitted pulse. Considering an incident Gaussian wave pulse with the central frequency $\omega_0$ and central wavevector component $k_{0x}$, the transmitted wave packet will be a STVP with the phase vortex in the $(x, t)$ plane and winding number opposite to that of the transmission-spectrum vortex (see the derivation in Supplementary Note 2). Because of the topological robustness of vortices with $l = \pm1$ in the $(k_x, \omega)$ plane or the nodal line in the $(k_x, \omega, \eta)$ space, a small perturbation of the system can only slightly move but not eliminate or create these entities. Away from the critical point $\eta = \eta_c$, small changes of the meta-grating geometry can be treated as perturbations. The strength of the topological protection for the vortex at $(k_{0x}, \omega_0)$ can be quantified by the distance to the nearest vortex: $\Delta = \sqrt{(\omega_1 - \omega_0)^2 + v^2(k_{1x} - k_{0x})^2}$, where $\omega_1$ and $k_{1x}$ are the frequency and the wavevector component of the nearest vortex, whereas $v$ is the speed of sound.

We now demonstrate the topologically protected generation of acoustic STVPs, schematically displayed in Fig. 3a. We choose the meta-grating with the asymmetry parameter $\eta = 1.0$, which has a relatively strong topological protection $\Delta \approx 14.4$ kHz (see Supplementary

Fig. S5). The measured transmission spectrum function exhibits a vortex at $\omega_0/2\pi = 8.02$ kHz and $k_{0x} = 0.01k_0$, where $k_0 = \omega_0/v$ is the wavenumber in air (see Supplementary Note 3 and Supplementary Fig. S6). Since $k_{0x} \ll k_0$, these parameters correspond to a near-normally incident (i.e., $z$-propagating) pulse. Using an arc-like linear transducer array, with an oscillatory Gaussian-envelope electric signal with the carrier frequency $\omega_0$, we produce $z$-propagating Gaussian pulses which have the full waist of 145 mm, the duration of 3.2 ms, and the diffraction Rayleigh range about 383 mm. For details of the experimental setup and measurements see Methods and Supplementary Fig. S8.

We first use the unperturbed meta-grating shown in Fig. 3b. We numerically simulate and experimentally measure the complex envelope $S_{out}(x,t)$ of the pressure field of the transmitted pulse $P_{out}(x,t) = S_{out}(x,t)e^{-i\omega_0 t}$ at different $z$-propagation distances of 64.4, 78.8, 93.2, and 107.6 mm, which are separated by the $\lambda_0/3$ intervals ($\lambda_0 = 2\pi/k_0$ is the center wavelength of the pulse) as indicated by red dash lines in Fig. 3a. These numerical and experimental results are depicted in Fig. 3d–g and Fig. 3h–k, respectively. One can clearly see the phase vortex with topological charge $l = -1$ in the spacetime $(x,t)$ domain. The phase rotates around the central nodal point upon the $z$-propagation of the pulse with the central spatial frequency $k_0$. Thus, the transmitted pulse is a first-order STVP.

Using the experimental data and numerical $z$-propagation of the field, we also calculate the spatial distributions of the transmitted-pulse amplitude $|P_{out}(x,z)|$ and the acoustic momentum density $\boldsymbol{\Pi} \propto \text{Im}[P_{out}^*(x,z)\boldsymbol{\nabla}P_{out}(x,z)]$ at different instants of time $t$, Fig. 4. These distributions show the propagation evolution and diffraction of the generated STVP. One can see a doughnut-like spatial shape, distorted by the diffraction but having topologically-protected intensity zero in the center (see Fig. 4c, d). Accurate calculations of the intrinsic OAM carried by the acoustic STVP are beyond the scope of this work, because there is an ongoing theoretical debate about the accurate definition of this quantity[33,35,39]. Nonetheless, Supplementary Note 4 shows that different equations for the intrinsic OAM yield nearly the same value of $L_y \approx 4$ in the units $\hbar$ per phonon because of the strong elongation of the doughnut-like pulse shape along the $z$-direction.

Finally, we demonstrate the topological robustness of our method of the STVP generation by perturbing the meta-grating structure. Namely, we randomly place 16 photopolymer-resins particles of different shapes and sizes about 0.8–1 cm, as shown in Fig. 3c. Moreover, as a regular perturbation of the grating, we also add one more block in the unit cell. The transmission spectrum of the perturbed meta-grating still exhibits the same phase singularity with a slightly shifted position

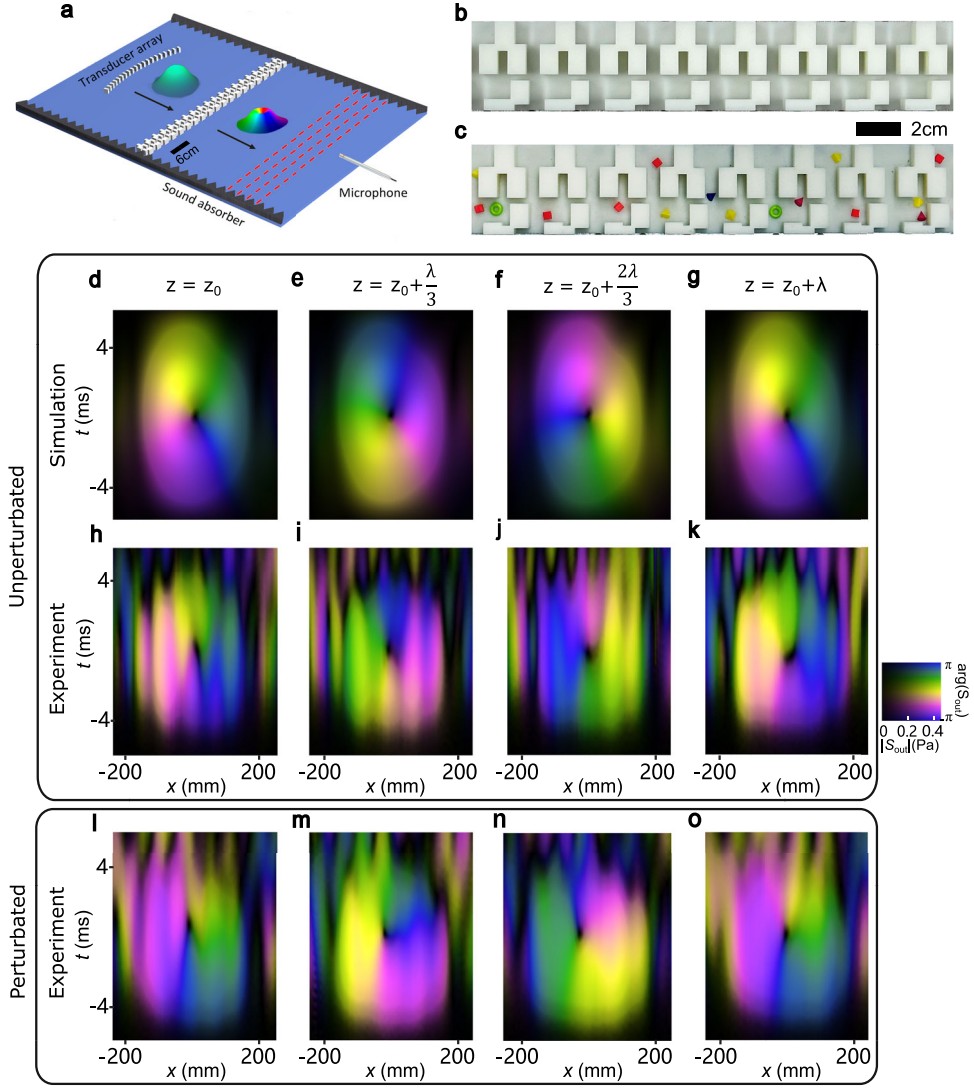

**Fig. 3 | Topologically protected generation of acoustic spatiotemporal vortex pulses. a** Schematics of the experimental setup with the incident acoustic Gaussian pulse and transmitted acoustic STVP. **b** Experimental sample of the acoustic meta-grating with the asymmetry parameter $\eta = 1.0$. **c** The perturbed meta-grating where 16 particles of different shapes (sphere, pyramid, cube and ring) are randomly placed, and additional regular small cuts are introduced. **d–g** Numerical simulations of the transmitted pulse envelopes $S_{out}(x,t)$ in the space-time domain at different $z$-positions separated by the $\lambda_0/3$ intervals ($\lambda_0$ is the central wavelength of the pulse). **h–k** Experimental measurements of the transmitted pulse envelopes corresponding to the numerical simulations in (**d–g**). **l–o** Experimental measurements of the topologically protected STVP generation using the perturbed meta-grating shown in (**c**).

$\omega_0/2\pi = 7.56$ kHz and $k_{x0} = 0.02 k_0$ (see Supplementary Fig. S7). Adjusting the central frequency of the incident pulse to this perturbed $\omega_0$, we measure the transmitted pulse envelopes at the corresponding $\lambda_0/3$-separated propagation distances of 65.3, 80.6, 95.9, and 111.2 mm. Figure 3l–o clearly show the transmitted STVP quite similar to that in the unperturbed case, Fig. 3h–k. We also numerically simulated the transmitted STVP for the perturbated meta-grating (Supplementary Fig. S9).

## Discussion

We have demonstrated the topologically protected generation of acoustic STVPs, carrying phase vortices in two spatial and one temporal dimensions, using a 1D periodic meta-grating with broken mirror symmetry. On the one hand, acoustic STVPs open an avenue for acoustic spacetime-structured waves, so far mostly studied in optics[40,41]. On the other hand, our method of STVP generation can find applications in acoustics, optics, and for other types of waves. One can expect that by designing 2D metasurfaces with an additional spatial dimension, one can synthesize $(3+1)$D spatiotemporal

vortices, such as vortices with arbitrarily tilted OAM[30,31]. In general, due to the geometric and physical differences from monochromatic vortex beams, STVPs can bring novel functionalities to acoustic/optical manipulation of particles, information transfer, and other applications[7,11,32,51–55].

We also note that the vortex in the momentum-frequency domain of the transmission spectrum function offers a new way to control the wave flow. In particular, the amplitude of the transmission spectrum function exhibits a linear dependence near the vortex center. Therefore, similarly to the image processing and edge detection in the spatial domain[56–68], the STVP generator can be treated as a first-order *differentiator* producing the derivative of the envelope in both spatial and temporal domains. This allows one to extract information about the space-time boundary of the incident sound, which can have applications in sonar and sensing. Moreover, acoustic vortices and generally spacetime-structured waves can be highly important for acoustic and acoustofluidic applications for the manipulation of bio-medical objects (cells and microorganisms), while optical structured waves are not sufficient for this[69].

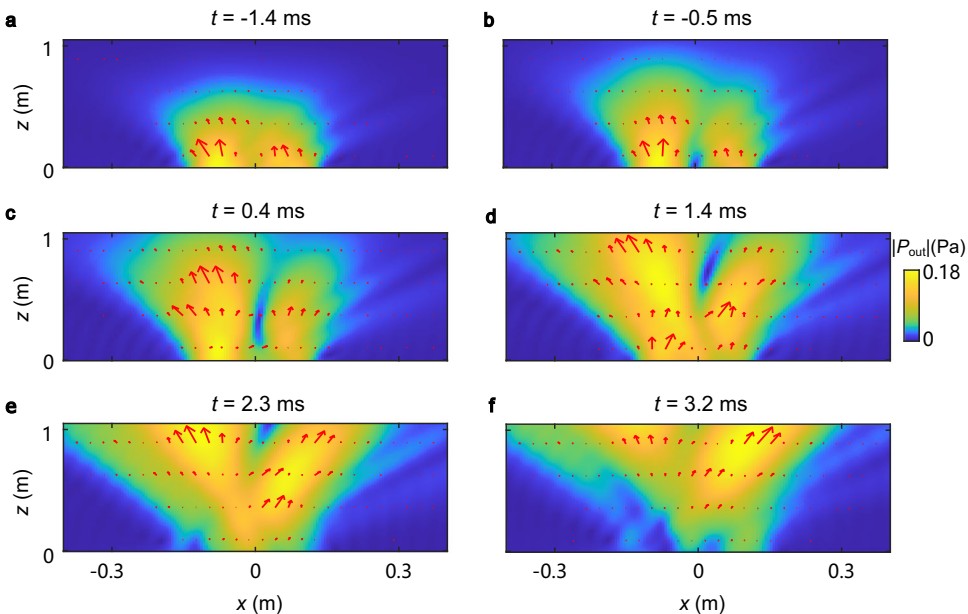

**Fig. 4 | Propagation of acoustic spatiotemporal vortex pulse in real space.**
**a–f** Spatial distributions of the pressure field amplitude, $|P_{out}(x,z)|$ (colormap), and the wave momentum density $\boldsymbol{\Pi}(x,z)$ (arrows) in the transmit*t*ed STVP at different instants of time $t$. One can see the front edge of the pulse (**a**, **b**), the nodal point in the center (**c**, **d**), and the rear edge of the pulse (**e**, **f**). The pulse shape is deformed due to the diffraction.

Note added in the proof: While our paper was in review, another paper was published in Phys. Rev. Lett., which also studies the generation of acoustic spatiotemporal vortices[70].

## Methods
### Fabrication and simulation of acoustic meta-grating
We used commercial services to fabricate the samples, with a 3D printing method, stereolithography, by using a laser to selectively solidify liquid photopolymer resin, layer by layer, to create solid objects. The transmission spectrum functions are numerically simulated by the finite-element method. Since the impedance of epoxy resin materials is much higher than that of air, the interface was regarded as a hard boundary. We simulate the transmitted pulses by the Fourier transform method with the transmission spectrum function and check the convergence of results.

### Experimental setup and methods to measure the transmission spectrum function
The experimental setup is shown in Fig. S3(a). A data acquisition (Brüel & Kjær 3160-A-042-R) is used to collect the data of acoustic field and control the output waveform. Two microphones (Brüel & Kjær 4193-L-004) are connected to the data acquisition and used to measure the acoustic field. We use a power amplifier (Brüel & Kjær 2735) to amplify the input signal. The displacement platform (LINBOU NFS03) and the data acquisition are integrated into a PC. The meta-grating and the sound absorber in Fig. 3a are placed between two glass plates. Thus, we deal with a quasi-2D (x,z) system similar to a planar waveguide between the two glass plates.

To measure the transmission spectrum function, an array consisting of ten transducers, with the distance about 1 cm from each other, is utilized to generate an incident field with sufficiently wide spatial spectrum range which does not overlap with higher diffraction orders, Fig. S3(b). Furthermore, a series of pulses is generated with the frequency spectrum ranging from 7 to 9 kHz.

Furthermore, we use a pair of microphones, with one designated as the reference and the other as the probe. The reference microphone is rigidly positioned at a fixed location within the acoustic field, while the probe microphone scans along the x-direction. At each probe position, the reference and probe microphones record time signals concurrently. By comparing the time signal detected by the probe with the reference signal, the precise time reaching the probe at each measurement point is measured by aligning all the reference signals. In doing so, we first measure the incident acoustic wave $P_{in}(x,t)$ at the scanning line (Fig. S3) without a meta-grating. Then the transmitted acoustic wave $P_{out}(x,t)$ is measured in the presence of the meta-grating. By applying the space-time Fourier transform to the incident (transmitted) acoustic field, we obtain the complex amplitudes of the incident (transmitted) plane waves. This provides the data for the transmission spectral function.

### Experimental setup and measurement principle of STVP
We use a curved transducer array to simultaneously generate a series of pulses with a Gaussian envelope at the central frequency $\omega_0$, as shown in Fig. S8(a). The spatial shape of the curved transducer array is described by the function $z = \exp(x^2/\sigma^2) - 1$, whereas the x-interval between the transducers is 1 cm. We put the meta-grating at the distance of 50 cm from the transducer array. The pair-microphones method, which is used to measure the transmission spectrum function, is also applied to measure the STVPs. First, the incident wave $P_{in}(x,t) = S_{in}(x,t)e^{-i\omega_0 t}$ is measured, with the envelope distribution $S_{in}(x,t)$ shown in Figs. S8(b–d). Then, the transmitted wave $P_{out}(x,t) = S_{out}(x,t)e^{-i\omega_0 t}$ is measured by scanning the field at different z-positions.

## Data availability
The data that support the findings of this study are available from the corresponding author upon request.

## Code availability
The code in this paper are available from the corresponding author upon request.

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

## Acknowledgements

The authors acknowledge funding through the National Key Research and Development Program of China (Grant No. 2022YFA1405200, 2022YFA1404203), the National Natural Science Foundation of China (NSFC Grants Nos. 12174340, 12174339). Z.Y. acknowledges Zhejiang Provincial Natural Science Foundation of China under Grant No. LR23A040003, and the Excellent Youth Science Foundation Project (Overseas). K.Y.B. thanks ENSEMBLE3 Project (MAB/2020/14) which is carried out within the International Research Agendas Programme (IRAP) of the Foundation for Polish Science co-financed by the European Union under the European Regional Development Fund and Teaming Horizon 2020 programme of the European Commission; the TEAM/2016-3/29 Grant within the TEAM program of the Foundation for Polish Science co-financed by the European Union under the European Regional Development Fund.

## Author contributions

Z.R. initiated the idea and this project. H.Z., Y.S., and J.H. developed the meta-grating design, and performed the experiments and measurements. J.H. and B.W. performed numerical calculations of angular momentum. Z.R., Z.Y., and K.Y.B. analyzed the experimental data and wrote the manuscript. Z.R. and Z.Y. supervised the project.

## Competing interests

Z.R., Z.Y., H.Z., Y.S., J.H., and B.W. are named inventors on a number of patent applications related to this work. The other authors of this work declare no other competing interests.
