## [Peer Review File · Nature Communications]

Observation of acoustic spatiotemporal vorticesREVIEWER COMMENTS

Reviewer #1 (Remarks to the Author):

Zhang et al describe the experimental generation and detection of spatiotemporal vortices using acoustic fields, which until now have been explored within optics. The vortices are constructed within a synthetic dimensional space comprising frequency, transverse wavenumber, and structure component displacement. The vortices are generated with a curved transducer array and an acoustic metasurface and are detected in the time/transverse position coordinate system. The robustness of the vortex existence is evaluated using randomly placed scatterers within the metasurface.

While I appreciate the novelty of the paper in the sense that it appears to be the first to demonstrate an experimental realization of acoustic spatiotemporal vortices, I am unable to endorse it for publication for the following reasons: 1. There are no new physical principles, design principles, or experimental protocols introduced in this study, and 2. The results are ambiguous at best due to truncated axes and colormaps, and missing data and protocols.

The extension of spatiotemporal vortices to acoustic systems has been examined theoretically by one of the study's authors (K.Y. Bliokh, "Orbital angular momentum of optical, acoustic, and quantum-mechanical spatiotemporal vortex pulses", Phys Rev A 2023). It is indeed a nontrivial feat to experimentally realize any phenomenon that has been predicted numerically, but in this case, the authors need to substantially strengthen their paper by including generalizable insights such as observations or explanations of new physical behavior, techniques or design principles for designing such vortices - there is little for future studies to build on top of this report. In particular, the paper stands to benefit from a discussion of metasurface design apart from a weak condition that it has to be parity asymmetric.

There are further discrepancies in the experimental results which render the conclusions unconvincing. In particular, the inconsistent transverse wavenumber axes in Fig 1-2 and unreported colorbar labels in Fig 2-4 do not support the manuscript argument (about the sufficiency of structure asymmetry, existence of a zero-intensity position) well. Full details are provided in the page-wise breakdown below.

After significant revisions, this manuscript may be suitable for publication in a discipline-specific journal. Unfortunately, its narrow scope and lack of general applicability make it less of a candidate for the multidisciplinary audience of Nature Communications.

Nevertheless, I would like to suggest the following manuscript changes for the authors' consideration in future submissions:

- All pages: Please pick a coordinate system (k, ω) or (ω, k) and be consistent throughout.
- Page 1: Note that vortices do not always transmit OAM (see Berry and Liu, "No general relation between phase vortices and orbital angular momentum", J Phys A, 2022).
- Page 2: Why is it necessary to include the scaling parameter A_i ? This makes it very difficult to interpret the physical significance of η . There is no justification for the choice of each A_i in the Supplementary.

- Page 3: Please justify the designation of the bending of the nodal line as a topological phase transition. The rotation sense of the phase about the nodal line remains the same in both sections and the only difference is that the nodal line direction into the eta plane is opposite in both charges. The bending of the nodal line is not unique to this asymmetric platform and occurs in random speckle (O'Holleran and Padgett, "Topology of light's darkness", PRL 2009) - does that mean that any tangent plane to the bend produces a topological phase transition as well?
- Page 4: Please specify the range of incident transverse wavenumbers corresponding to the transducer and metasurface geometry.
- Figure 1: Periodicity p is not defined in the figure or caption, or main text. Please label it in Figure 1a.
- Figure 1b: Please expand the horizontal k_x axis of this structure to include $k_x * p = \text{plus minus } 0.25$ to match Figures 1c-d and make a proper comparison. The asymmetric metasurfaces exhibit the singularity at much larger $k_x * p$ values than plus/minus 0.25 and so it is important to have a direct comparison between the transmission behavior at these tilt angles for the symmetric and asymmetric cases.
- Figure 1c-d: Please plot the negative k_x values for these too to demonstrate the degree of asymmetric transmission. This is important since you are using a transducer array that is symmetric about the axis and thus introduces both positive and negative wavenumbers (Fig 3a).
- Figure 2 and 3l-o and 4: Please provide quantitative values for the transmission amplitude, and not just "max" and "min". Please set the minimum intensity on the colorbar to zero or use logarithmic scaling as done in Fig 1. What is the contrast between the maximum and minimum intensities obtained? How do you know that the value at the center is identically zero less equipment noise?
- Figure 2: Please report the symmetric transmission function over a wider range of $k_x * p$ values at least up to $k_x * p = 1.5$ to match that of the bottom right plot. Please compare the phase and amplitude plots obtained experimentally and explain the significant differences with the numerically obtained plots in Figure 1b-d. Do the experimentally-obtained vortices in Fig 2 correspond to the vortices plotted in Figure 1b-d? If so, why are their k_x coordinates off by a factor of 2 or more?
- Figure 3. Please explain why the x - t vortex appears much cleaner and symmetric than the k - ω vortex in Fig S4a-b. Does the $\eta=1$ metasurface have vortices away from the axis (i.e. when $|k_x|/k_0 > 0.05$)? Why is there only one vortex in the $\eta=1$ metasurface when previous metasurfaces (like the $\eta=0.5, 0.7$ metasurfaces) have more than one vortex in k - ω space?
- Page 6 methods: definition of the Gaussian is missing a sign.
- Methods: Please include a section for the fabrication, materials, and material properties used in your metasurface and in simulating the transmission response.
- Page 4: You show that a vortex in k - ω space will produce a vortex in the time/transverse space you use in the experiment. Is the converse true: does the observation of a vortex in the time/transverse space imply a vortex in k - ω space?
- Supplementary S1 needs to be substantially expanded and supported mathematically. It is currently a series of assertions instead of a proof.
- Supplementary S2: Please check the dimensions of the variables used - x and t (and K_x and Ω) seem to be assigned the same units. If you have non-dimensionalized your parameters, please state the scaling factors.

Reviewer #2 (Remarks to the Author):

In this work, the authors design and experimentally demonstrate a metasurface for creating acoustic spatiotemporal vortices. The ability to control spatiotemporal vortices has fostered strong interest in the field of optics, this expansion to acoustics has the potential to provide new means of acoustic control. With appropriate revisions, I think this paper is appropriate for publication in nature communications.

Major:

1. Can the authors comment on the advantages of acoustic vortices for sensing applications?
2. What was the heuristic for the design of the metasurface? Is it resonant or non-resonant? From the transmission plots in the SI, it seems non-resonant but this may be the result of an experimental limitation.
3. In Figures 1 & 2, why do the formation of the of vortices appear at some arbitrary value of η ? What is the origin of formation for the vortices so that they form at a critical value of η ?
4. In figure 3, the authors present t-x slices showing acoustic vortices. What do these plots look like in the planes perpendicular to the direction of propagation? Do the vortices only exist in the t-x domain?
5. On page 5, there is a paragraph discussing results of spatial differentiation. As it stands, this section is confusing in the discussion session. There is too little information to contextualize the point. Along these lines, does spatial differentiation require the acoustic vortices? From the figure in the SI, the differentiation only needs a k-vector dependent transmission profile to see the effect.

Reviewer #3 (Remarks to the Author):

Spatiotemporal vortices in optics has been endorsed as a hot topic in recent years. Now the authors experimentally generated spatiotemporal vortices in sound waves, which is very timely physical breakthrough. For the novelty, it can fulfill the criteria of nature communications. While, the current version includes several unclear or confusing statements to be addressed before considering publication.

1. The authors use terminology of topological texture several times to describe spatiotemporal vortices. I don't think it is suitable, as topological texture is a terminology widely used in spintronics and skyrmions, which refers to solitonic configuration in spin vector patterns. However, spatiotemporal vortices are purely scalar light fields with phase singularities, there is no vector texture.
2. the authors highlighted flying doughnut-shaped pulses, but I am very shocked that the authors neglected the typical works on flying electromagnetic doughnut or toroidal light pulse for both theoretical and experimental generation works [Nat Commun 12, 5891 (2021); Nat Photon 16, 523–528 (2022)]
3. the generated acoustic spatiotemporal vortex pulse should be propagating in free space. But the setup figure (Fig.3a) likely draw surface waves propagating on a surface, which will easily introduce confusion.
4. About the topological stability in propagation. In my opinion, spatiotemporal vortices are theoretically not propagating stable modes, the patterns are changing upon propagation (not like LG

modes). If the author send a high topological charge spatiotemporal vortex, the stability cannot be observed I think. Therefore the reason for the stability is not for spatiotemporal vortices, but the generic singularity (the authors only demonstrate the generic case).

5. Also, in Fig.3, the simulation results for the perturbed spatiotemporal vortex evolution should also be provided.

6. I personally understand the color used for plots Figs 3d-3o, but I have to say general readers are easily confused by the colorbar, "there is a gray colorbar but no gray color in the plots???" so my suggestion is to provide a 2D colorbar including informations of both intensity represented by lightness and phase by hue color.

7. The relationship between k- ω domain and x-t domain topological charge has been introduced in this paper, and k- ω domain vortices are realized by asymmetric meta-gratings, and then x-t domain vortices are generated. However, the relationship between asymmetric meta-gratings and K- ω -domain vortices remains unclear. Can any asymmetric meta-gratings generate k- ω -domain vortices?

8. The linear response of transmission amplitude T in the k- ω domain and the inverse response of transmission phase are shown in the supplementary material, which is the classical differentiator response. It is necessary to clarify whether there is a necessary relationship between spatiotemporal vortices response and differentiator response. Is the transmission amplitude T at the vortex in the k- ω domain necessarily linear?

9. The measuring method of X-T domain needs to be introduced with more details. The probe can only test the time-domain signal of one position x each time, right? How to calibrate t of different x positions to splicing X-T results?

Reviewer #4 (Remarks to the Author):

The authors have submitted their work "Observation of acoustic spatiotemporal vortices" for consideration to be published in Nature Communications.

I have read the work in detail, including the supplementary information, and I think that it can be published in Nature Communications. However, I take issue mainly with a few organizational choices made by the authors. Mainly that a large number of details are included only in the supplementary document, which I think are crucial to include in the main paper, or that I think there should be more detail in general. I understand that many of these comments may be partially opinion-based or an editorial decision that is related to the journal, but I write them in the spirit of improving the paper and making it more useful for readers. In general, the paper is well-written and the figures in the main article are of high quality.

I will number a few specific and general comment below:

1. The parameter η is the crucial design parameter for the grating structure, but it is only truly defined in the SM. Since it is only there, the writing and organization of the definition and its different components (text, figure, numbers, etc.) is not ideal. Because of this it took quite some time to find how this was defined, and the presentation still makes it less transparent than it should be.

2. The grating structure is obscured in Fig. 1a and too small in the other sub-figures. It is very clear in Fig. 3b, but it would be more sensible if it were clear at the beginning of the main article.

3. You do not even mention briefly in the main text how you measured the transmission function. Yes, it is in the methods, but that should be for details. The text should still have some information.

a. In the methods, you write “To measure the transmission spectrum function in the frequency domain (as shown in Fig. S2(b)), we use 10 transducers to form a rectangle source in the spatial domain, which ensures that the spatial spectrum of the incident field is sufficiently wide but don’t overlap with the nonzero diffraction order.” I don’t see such detailed information about the source in Fig. S2(b), where there is only a point source. And in any case, it seems like there should be more detail on this point.

b. Again, since these details are saved for the SM, the figures and the explanations are of lower quality, which I think is very unfortunate for the reader.

4. The above also applies to the experimental setup to measure the STVP, albeit less so since there is a sketch in Fig. 3a, and Fig. S6 is of higher quality.

5. General comment: Very little of the work focuses on the fact that these are acoustic waves (longitudinal pressure waves) and not transverse electromagnetic waves. This is present in Figure 4, but it is not clear what implication this had on the design of the meta-grating and the simulations necessary to propagate the waves, etc. It would have been nice for there to be more detail on this aspect in the early part of the main article, to set the stage for the work and to provide more context and detail. Are you relying too much on Ref. [33] for that?

We thank the Referees for careful reading of our manuscript and relevant remarks. We have addressed these remarks in the revised version of the manuscript and additionally polished its text to improve its clarity. Below we respond point by point to the Referees' comments.

REFeree 1: *While I appreciate the novelty of the paper in the sense that it appears to be the first to demonstrate an experimental realization of acoustic spatiotemporal vortices, I am unable to endorse it for publication for the following reasons: 1. There are no new physical principles, design principles, or experimental protocols introduced in this study, and 2. The results are ambiguous at best due to truncated axes and colormaps, and missing data and protocols.*

The extension of spatiotemporal vortices to acoustic systems has been examined theoretically by one of the study's authors (K.Y. Bliokh, "Orbital angular momentum of optical, acoustic, and quantum-mechanical spatiotemporal vortex pulses", Phys Rev A 2023). It is indeed a nontrivial feat to experimentally realize any phenomenon that has been predicted numerically, but in this case, the authors need to substantially strengthen their paper by including generalizable insights such as observations or explanations of new physical behavior, techniques or design principles for designing such vortices - there is little for future studies to build on top of this report. In particular, the paper stands to benefit from a discussion of metasurface design apart from a weak condition that it has to be parity asymmetric.

There are further discrepancies in the experimental results which render the conclusions unconvincing. In particular, the inconsistent transverse wavenumber axes in Fig 1-2 and unreported colorbar labels in Fig 2-4 do not support the manuscript argument (about the sufficiency of structure asymmetry, existence of a zero-intensity position) well. Full details are provided in the page-wise breakdown below.

After significant revisions, this manuscript may be suitable for publication in a discipline-specific journal. Unfortunately, its narrow scope and lack of general applicability make it less of a candidate for the multidisciplinary audience of Nature Communications.

OUR RESPONSE: We thank the Referee for the comments which have helped us to improve the manuscript.

Our work has a two-fold novelty. First, we report the first experimental generation of acoustic spatiotemporal vortex pulses (STVPs). Such pulses have indeed been mentioned theoretically in [33], but this is not a big deal and not the central result of [33] because STVPs are universal wave solutions present in any linear wave equation. The real challenge is to generate such states experimentally. In optics, such states were generated several years after their theoretical prediction [25], and these results were published in high-impact journals [26-28].

Second, for the generation of acoustic STVPs, we use a totally novel principle and experimental design. Instead of cylindrical lenses and phase plates used in optical works [27,28], we use a meta-grating with nontrivial topological properties (phase singularities) of the transmission

spectrum function. This method was previously suggested theoretically for optical systems [31,32] but in our work, it has been implemented experimentally for the first time. Importantly, this method can be applied to different types of waves, and it has significant advantages: (i) it uses a single narrow meta-grating/surface instead of extended lens systems and (ii) it is topologically robust with respect to the disorder in the system.

Concerning the “ambiguity” of our results, our experimental measurements and numerical simulations present perfectly clear vortices in the space-time domain, which cannot be confused with anything else because of the topological nature of the phase winding in this object. One can easily check that our experimental data are as clear as any previous results about optical STVPs. In the revised version of this manuscript, we did our best to provide all the additional information requested by the Referees.

Therefore, we believe that our work provides considerable progress in studies and generation of STVPs (in acoustics and beyond), and that it will attract great interest of the multidisciplinary readership of Nature Communications. Furthermore, spacetime-structured acoustic waves can be highly important for acoustic and acoustofluidic applications for the manipulation of biomedical objects (cells and microorganisms) , see Ref. [68], while optical structured waves are not sufficient for this.

REFEREE 1: *All pages: Please pick a coordinate system (k, ω) or (ω, k) and be consistent throughout.*

OUR RESPONSE: We have unified the coordinate system (k, ω) across the paper.

REFEREE 1: *Page 1: Note that vortices do not always transmit OAM (see Berry and Liu, “No general relation between phase vortices and orbital angular momentum”, J Phys A, 2022).*

OUR RESPONSE: We are aware of this interesting work and certainly agree that there is no direct relation between the phase vortices and OAM. In fact, in our study, we only report the generation of STVPs but do not study their OAM. We only refer to previously published results about the OAM of STVPs (in the Introduction) and estimate the OAM of STVPs generated in our experiment using previously published theoretical calculations [33,35,39]. We remark in the text that accurate calculations of the intrinsic OAM carried by acoustic STVPs are beyond the scope of our study and it requires more detailed theoretical and experimental investigation.

REFEREE 1: *Page 2: Why is it necessary to include the scaling parameter A_i ? This makes it very difficult to interpret the physical significance of η . There is no justification for the choice of each A_i in the Supplementary.*

OUR RESPONSE: The purpose of introducing the scaling parameters A_i is to describe the degree of the mirror symmetry breaking by a single dimensionless parameter η . The dimensional coefficients A_i describe the characteristic shift magnitudes and directions for each block, while η provides the overall multiplication factor for these shifts. Therefore, this is a synthetic parameter characterizing the degree of asymmetry, where $\eta = 0$ ($\eta \neq 0$) corresponds to the cases where the

four blocks are mirror symmetric (asymmetric). Based on this idea, we numerically simulate different meta-gratings and choose A_i for each block to obtain the meta-grating with desired transmission properties.

REFeree 1: *Page 3: Please justify the designation of the bending of the nodal line as a topological phase transition. The rotation sense of the phase about the nodal line remains the same in both sections and the only difference is that the nodal line direction into the eta plane is opposite in both charges. The bending of the nodal line is not unique to this asymmetric platform and occurs in random speckle (O'Holleran and Padgett, "Topology of light's darkness", PRL 2009) - does that mean that any tangent plane to the bend produces a topological phase transition as well?*

OUR RESPONSE: We agree with the Referee that the topology of nodal lines does not uniquely exist in our design structure, it is a universal feature. By topological transition we mean not merely bending of the nodal line in 3D space but rather the event when the selected 2D plane touches the nodal line. This is a point of topological transition for the function in the 2D plane, (k_x, ω) in our case. Indeed, below the critical value, $\eta < \eta_c$, there are no phase singularities in $T(k_x, \omega)$, whereas a pair of opposite-sign singularities appear for $\eta > \eta_c$. Therefore, we call this "topological phase transition" for the function $T(k_x, \omega)$ at the critical value of the parameter $\eta = \eta_c$. It should be emphasized that the same behavior is a topological transition in the 2D plane (k_x, ω) but just a single nodal line with topology in the extended 3D space (k_x, ω, η) . Such dependence of topological features on the dimensionality of the space under consideration is well known, see e.g. Ref. [48, 49]. We have revised the description of these features in our manuscript to make this clearer.

[48]O'holleran, K., Dennis, M. R. & Padgett, M. J. Topology of light's darkness. Phys. Rev. Lett. 102, 143902 (2009).

[49]Berry, M. V. Much ado about nothing: optical distortion lines (phase singularities, zeros, and vortices). SPIE. 1-5(1998).

REFeree 1: *Page 4: Please specify the range of incident transverse wavenumbers corresponding to the transducer and metasurface geometry.*

OUR RESPONSE: We used incident Gaussian pulses with the full waist of 145 mm (see Supplementary Fig. S6). This corresponds to the Gaussian profile of the incident transverse wavenumbers with the half waist of $0.2k_0$.

REFeree 1: *Figure 1: Periodicity p is not defined in the figure or caption, or main text. Please label it in Figure 1a.*

OUR RESPONSE: We have added the definition of p in the revised Fig. 1.

REFeree 1: **Figure 1b:** *Please expand the horizontal kx axis of this structure to include $kx*p = \text{plus minus } 0.25$ to match Figures 1c-d and make a proper comparison. The asymmetric metasurfaces exhibit the singularity at much larger $kx*p$ values than plus/minus 0.25 and so it is important to have a direct comparison between the transmission behavior at these tilt angles for the*

symmetric and asymmetric cases.

Figure 1c-d: *Please plot the negative k_x values for these too to demonstrate the degree of asymmetric transmission. This is important since you are using a transducer array that is symmetric about the axis and thus introduces both positive and negative wavenumbers (Fig 3a).*

OUR RESPONSE: We have revised Figs. 1b-d as requested by the Referee.

REFeree 1: *Figure 2 and 3l-o and 4: Please provide quantitative values for the transmission amplitude, and not just “max” and “min”. Please set the minimum intensity on the colorbar to zero or use logarithmic scaling as done in Fig 1. What is the contrast between the maximum and minimum intensities obtained? How do you know that the value at the center is identically zero less equipment noise?*

OUR RESPONSE: We have revised the color bars in Figs. 2, 3l-o, and 4 as suggested by the Referee. The zero of intensity is guaranteed by the measured phase singularity in the vortex. As soon as there is a $\pm 2\pi$ phase increment around the vortex center, there is a phase singularity point with zero intensity (phase is undefined only for zero or infinite amplitude). Its exact location can be limited by the experimental noise but its presence is a rigorous topological property of any smooth complex function.

REFeree 1: *Figure 2: Please report the symmetric transmission function over a wider range of $k_x \cdot p$ values at least up to $k_x \cdot p = 1.5$ to match that of the bottom right plot. Please compare the phase and amplitude plots obtained experimentally and explain the significant differences with the numerically obtained plots in Figure 1b-d. Do the experimentally-obtained vortices in Fig 2 correspond to the vortices plotted in Figure 1b-d? If so, why are their k_x coordinates off by a factor of 2 or more?*

OUR RESPONSE: We thank the Referee for noticing this discrepancy. There was a typo in Figure 2 where the scale of 10^{-1} was omitted in the horizontal axes, which resulted in the exaggerated deviation between the experimental and numerical results. We have corrected this typo in the revised manuscript.

The measurable transverse wavevector range is limited by the experimental incident signal, which is a Gaussian beam with a limited k_x range. For this reason, in order to cover the two vortices in the transmission spectrum, the experimental data for $\eta = 0.7$ were measured with two different incident angles, see Figs. 2c and f. Figure R1 below shows the same simulation data as in Fig. 1c-e but with the (k_x, ω) ranges appropriate for the comparison with the experimental results of Fig. 2. One can see a very good agreement between the numerical and experimental results.

Fig. R1. Simulation data of Fig. 1c-e for the comparison with the experimental results of Fig. 2.

REFEREE 1: *Figure 3. Please explain why the x - t vortex appears much cleaner and symmetric than the k - ω vortex in Fig S4a-b. Does the $\eta=1$ metasurface have vortices away from the axis (i.e. when $|k_x/k_0| > 0.05$)? Why is there only one vortex in the $\eta=1$ metasurface when previous metasurfaces (like the $\eta=0.5, 0.7$ metasurfaces) have more than one vortex in k - ω space?*

OUR RESPONSE: We think that the experimentally measured vortex in the (x, t) domain, Fig. 3h-k, has approximately the same level of perturbations as the vortex in the (k_x, ω) domain, Fig. S4. As long as the amplitude has a linear dependence near the nodal point, this can be considered as a good-quality generic vortex. This is the case in Fig. S4c,d.

The meta-surface with $\eta = 1$ also exhibits two vortices in the (k_x, ω) domain. These are shown in Fig. R2 below. We just focus on one of these vortices with the incident and transmitted pulses; the second vortex is far from the spectral parameters of our pulse.

Fig. R2. Numerical simulations for the transmission spectrum function of the meta-grating with $\eta = 1$.

REFEREE 1: *Page 6 methods: definition of the Gaussian is missing a sign.*

OUR RESPONSE: We thank the Referee for pointing to this inaccuracy. The position of the transducer array is not a Gaussian function, so we have removed the word ‘‘Gaussian’’ and kept the position profile as $z = \exp(x^2/\sigma^2) - 1$.

REFEREE 1: *Methods: Please include a section for the fabrication, materials, and material properties used in your metasurface and in simulating the transmission response.*

OUR RESPONSE: In accordance with the Referee's suggestion, we have added this information in the Method in the revised manuscript.

We used commercial services to fabricate the samples, with a 3D printing method, stereolithography, by using a laser to selectively solidify liquid photopolymer resin, layer by layer, to create solid objects. The transmission spectrum functions are numerically simulated by the finite-element method. Since the impedance of epoxy resin materials is much higher than that of air, the interface was regarded as a hard boundary. We simulate the transmitted pulses by the Fourier transform method with the transmission spectrum function and check the convergence of results.

REFEREE 1: *Supplementary S1 needs to be substantially expanded and supported mathematically. It is currently a series of assertions instead of a proof.*

OUR RESPONSE: Supplementary "S1. The details of geometry parameters of acoustic spatiotemporal vortex pulse generator" only shows geometric parameters of the meta-grating. This was not "derived" from any equations. The specific geometry and parameters of the metasurface are determined only by calculating the transmission spectrum function numerically and just following the symmetry breaking principle. We have substantially expanded this point in Supplementary S1.

REFEREE 1: *Supplementary S2: Please check the dimensions of the variables used - x and t (and K_x and Ω) seem to be assigned the same units. If you have non-dimensionalized your parameters, please state the scaling factors.*

OUR RESPONSE: We thank the Referee for pointing this out. We have added the corresponding scaling factor, which is the speed of sound.

=====

REFEREE 2: *In this work, the authors design and experimentally demonstrate a metasurface for creating acoustic spatiotemporal vortices. The ability to control spatiotemporal vortices has fostered strong interest in the field of optics, this expansion to acoustics has the potential to provide new means of acoustic control. With appropriate revisions, I think this paper is appropriate for publication in Nature Communications.*

OUR RESPONSE: We are grateful to the Referee for appreciation of our work and relevant comments, which have helped us to improve the manuscript.

REFEREE 2: *Can the authors comment on the advantages of acoustic vortices for sensing applications?*

OUR RESPONSE: In the Discussion section, we indicate that the proposed STVPs generator can be used to extract the space-time boundary information of the incident sound, i.e., the signal

differentiation. This can have applications in sonar and acoustic sensing. Moreover, acoustic vortices and generally spacetime-structured waves can be highly important for acoustic and acoustofluidic applications for the manipulation of biomedical objects (cells and microorganisms), see Ref. [68], while optical structured waves are not sufficient for this.

REFeree 2: *What was the heuristic for the design of the metasurface? Is it resonant or non-resonant? From the transmission plots in the SI, it seems non-resonant but this may be the result of an experimental limitation.*

OUR RESPONSE: As discussed in the main text “Subsec. Breaking spatial mirror symmetry for the generation of STVPs”, the metasurface is designed to create the phase vortices in the transmission spectrum, by breaking the mirror symmetry. Such a transmission spectrum feature is a non-resonant effect. In contrast to resonant systems, the transmission spectrum does not exhibit typical symmetric or anti-symmetric Lorentzian line shape neither. Moreover, the presence of phase singularities in the transmission spectrum is topologically robust with respect to the perturbations of the meta-surface structure, as discussed in the manuscript.

REFeree 2: *In Figures 1 & 2, why do the formation of the vortices appear at some arbitrary value of η ? What is the origin of formation for the vortices so that they form at a critical value of η ?*

OUR RESPONSE: This is because vortices do not exist in mirror-symmetric meta-surfaces with $\eta = 0$ and they can appear only in pairs as a result of “topological transition” when the (k_x, ω) plane touches the nodal line of the transmission spectrum function in the extended (k_x, ω, η) space. This happens at some critical value of the mirror-asymmetry parameter $\eta = \eta_c$, see Fig. 1. The shape of the nodal line and the critical value of this parameter depend on the specific geometry and parameters of the metasurface, and can be determined only by calculating the transmission spectrum function numerically.

REFeree 2: *In figure 3, the authors present t - x slices showing acoustic vortices. What do these plots look like in the planes perpendicular to the direction of propagation? Do the vortices only exist in the t - x domain?*

OUR RESPONSE: Here we only consider the spatiotemporal vortex propagating in the (x, z) , or (x, t) space-time domain, so that there is no special wave structure in the orthogonal y -direction. Therefore, in our experimental setup, the metasurface is positioned between two parallel glass plates $y=\text{const}$. The y -distance between these plates is less than $\lambda_0/2$, in order to ensure a near-uniform acoustic field along the y -direction.

REFeree 2: *On page 5, there is a paragraph discussing results of spatial differentiation. As it stands, this section is confusing in the discussion session. There is too little information to contextualize the point. Along these lines, does spatial differentiation require the acoustic vortices? From the figure in the SI, the differentiation only needs a k -vector dependent transmission profile to see the effect.*

OUR RESPONSE: Following the Referee's suggestion, we have revised this part in Discussion and now mention the spatiotemporal differentiation as a potential application for spacetime boundary detection and acoustic sensing. The main principle of the spatial and temporal differentiation has been described in detail in optical works [55-67], and it indeed only requires linear wavevector and frequency dependence of the transmission function, respectively. Such linear dependence is generically realized near phase singularities, so that vortices in the (k_x, ω) domain in our work can be used for spatiotemporal differentiation. These features do not use any specific properties of acoustic waves and can be equally realized in optics or for other types of waves.

=====

REFEREE 3: *Spatiotemporal vortices in optics has been endorsed as a hot topic in recent years. Now the authors experimentally generated spatiotemporal vortices in sound waves, which is very timely physical breakthrough. For the novelty, it can fulfill the criteria of Nature Communications. While, the current version includes several unclear or confusing statements to be addressed before considering publication.*

OUR RESPONSE: We are grateful to the Referee for appreciation of our work and relevant comments, which have helped us to improve the manuscript.

REFEREE 3: *The authors use terminology of topological texture several times to describe spatiotemporal vortices. I don't think it is suitable, as topological texture is a terminology widely used in spintronics and skyrmions, which refers to solitonic configuration in spin vector patterns. However, spatiotemporal vortices are purely scalar light fields with phase singularities, there is no vector texture.*

OUR RESPONSE: We agree with this point and removed the term “topological texture” from the manuscript.

REFEREE 3: *The authors highlighted flying doughnut-shaped pulses, but I am very shocked that the authors neglected the typical works on flying electromagnetic doughnut or toroidal light pulse for both theoretical and experimental generation works [Nat Commun 12, 5891 (2021); Nat Photon 16, 523–528 (2022)]*

OUR RESPONSE: We apologize for not mentioning works on flying electromagnetic doughnuts or toroidal light pulses. We have added the appropriate references [42-45] in the revised version of the manuscript.

[42] Shen, Y., Hou, Y., Papasimakis, N. & Zheludev, N. I. Supertoroidal light pulses as electromagnetic skyrmions propagating in free space. Nat. Commun. 12, 5891 (2021).

[43] Zdagkas, A. et al. Observation of toroidal pulses of light. Nat. Photonics 16, 523(2022).

REFEREE 3: *The generated acoustic spatiotemporal vortex pulse should be propagating in free space. But the setup figure (Fig.3a) likely draw surface waves propagating on a surface, which will easily introduce confusion.*

OUR RESPONSE: We appreciate the Referee's suggestion regarding Figure 3a. In the revision, we will modify Figure 3a to better represent the intended scenario of the acoustic spatiotemporal vortex pulse propagating in free space.

REFEREE 3: *About the topological stability in propagation. In my opinion, spatiotemporal vortices are theoretically not propagating stable modes, the patterns are changing upon propagation (not like LG modes). If the author send a high topological charge spatiotemporal vortex, the stability cannot be observed I think. Therefore the reason for the stability is not for spatiotemporal vortices, but the generic singularity (the authors only demonstrate the generic case).*

OUR RESPONSE: We thank the Referee for the opportunity to clarify some important points. In this work, we mention topological stability for the generation, not for propagation. Certainly, higher-order vortices are topologically unstable. In contrast, our generation method uses only the first-order phase singularity in the transmission spectrum function of the meta-grating. This first-order singularity is topologically stable with respect to any small perturbations of the meta-grating. Hence our STVP generator is topologically stable.

Note that the derivation in Supplementary Note 2 shows that when the input pulse has a spatiotemporal vortex with the winding number l , the output pulse will have the winding number of $l + 1$, even though the high-order spatiotemporal vortex could split into several lower-order ones.

REFEREE 3: *Also, in Fig.3, the simulation results for the perturbed spatiotemporal vortex evolution should also be provided.*

OUR RESPONSE: We have provided additional simulation results in Supplementary Fig. S7. Since simulating the whole irregularly perturbed structure is out of our computation resources, we only simulated a suitably perturbed periodic structure. Therefore, the central frequency of the pulse $\omega_0/2\pi=7.83\text{kHz}$ is different from the experimental one, but the generated spatiotemporal vortex is still clearly demonstrated.

REFEREE 3: *I personally understand the color used for plots Figs 3d-3o, but I have to say general readers are easily confused by the colorbar, "there is a gray colorbar but no gray color in the plots???" so my suggestion is to provide a 2D colorbar including informations of both intensity represented by lightness and phase by hue color.*

OUR RESPONSE: We thank the Referee for this suggestion. We have modified the colorbar in Figs. 3d-3o accordingly.

REFEREE 3: *The relationship between k - ω domain and x - t domain topological charge has been introduced in this paper, and k - ω domain vortices are realized by asymmetric meta-gratings, and then x - t domain vortices are generated. However, the relationship between asymmetric meta-gratings and K - ω -domain vortices remains unclear. Can any asymmetric meta-gratings generate k - ω -domain vortices?*

OUR RESPONSE: Although we show that the mirror-symmetry breaking *can* lead to the spectral vortex in the (k_x, ω) domain, this does not ensure that *any* asymmetric meta-gratings have such vortices. In particular, Fig. 1 shows that asymmetric meta-gratings with chosen design and $\eta < \eta_c$ do not exhibit phase singularities in the transmission spectrum function, whereas such gratings with $\eta > \eta_c$ have pairs of vortices in the (k_x, ω) domain.

REFeree 3: *The linear response of transmission amplitude T in the k - ω domain and the inverse response of transmission phase are shown in the supplementary material, which is the classical differentiator response. It is necessary to clarify whether there is a necessary relationship between spatiotemporal vortices response and differentiator response. Is the transmission amplitude T at the vortex in the k - ω domain necessarily linear?*

OUR RESPONSE: We think that there is no general relation between spatiotemporal vortices and differentiator response. In our proposed spatiotemporal vortex generator, the phase singularity in the transmission spectrum in the (k_x, ω) domain leads to the differentiator response. This is because the generic form of a first-order phase vortex assumes linear dependence of the amplitude on the distance from the nodal point. However, other methods for the generation of spatiotemporal vortices, e.g., the pulse shaping method, might not exhibit such transmission spectrum feature, and thus likely cannot perform differentiation. Also, the differentiator response does not necessarily require the spatiotemporal vortex generation, and there have been several proposals and systems to realize the spatial and temporal differentiation, Refs. [55-67], without involving spatiotemporal vortices.

For a vortex with the winding number of $l = \pm 1$, the transmission amplitude T in the k - ω domain is necessarily linear around the vortex. By decomposing a transmission spectrum with finite values by complete orthogonal basis of Hankel transformation $T(\rho, \theta) = \sum_{l=-\infty}^{\infty} e^{il\theta} \int A_l(k) J_l(k\rho) k dk$, where A_l and J_l are the coefficients and the l -th order Bessel function of the first kind. When $\rho \rightarrow 0$, $J_l(k\rho) \sim (k\rho)^{|l|}$ and thus $T \sim \rho e^{i\theta} + o(\rho)$, where $o(\rho)$ is higher order small quantities of ρ .

REFeree 3: *The measuring method of X-T domain needs to be introduced with more details. The probe can only test the time-domain signal of one position x each time, right? How to calibrate t of different x positions to splicing X-T results?*

OUR RESPONSE: We appreciate the Referee's suggestion and have added more details about the measurement of spatiotemporal vortices in the Methods.

In the experimental configuration, we utilize a pair of microphones, with one designated as the reference and the other as the probe, to accurately measure the acoustic fields. The reference microphone is rigidly positioned at a fixed location within the acoustic field, while the probe microphone scans along the x -direction. At each probe position, the reference and probe microphones record time signals concurrently. By comparing the time signal detected by the probe with the reference signal, the precise time reaching the probe at each measurement point is measured by aligning all the reference signals.

REFeree 4: *I have read the work in detail, including the supplementary information, and I think that it can be published in Nature Communications. However, I take issue mainly with a few organizational choices made by the authors. Mainly that a large number of details are included only in the supplementary document, which I think are crucial to include in the main paper, or that I think there should be more detail in general. I understand that many of these comments may be partially opinion-based or an editorial decision that is related to the journal, but I write them in the spirit of improving the paper and making it more useful for readers. In general, the paper is well-written and the figures in the main article are of high quality.*

OUR RESPONSE: We are grateful to the Referee for appreciation of our work and relevant comments, which have helped us to improve the manuscript.

REFeree 4: *The parameter η is the crucial design parameter for the grating structure, but it is only truly defined in the SM. Since it is only there, the writing and organization of the definition and its different components (text, figure, numbers, etc.) is not ideal. Because of this it took quite some time to find how this was defined, and the presentation still makes it less transparent than it should be.*

OUR RESPONSE: We appreciate and accept the Referee's suggestion. We have moved Fig. S1 of the initial version to Fig. 1b in the revised manuscript, such that the definition and presentation of the design parameter η are explicitly provided in the main text.

REFeree 4: *The grating structure is obscured in Fig. 1a and too small in the other sub-figures. It is very clear in Fig. 3b, but it would be more sensible if it were clear at the beginning of the main article.*

OUR RESPONSE: With the modification described in the previous comment, Fig. 1b also displays the grating structure in detail, and now it is clearly visible.

REFeree 4: *You do not even mention briefly in the main text how you measured the transmission function. Yes, it is in the methods, but that should be for details. The text should still have some information.*

a. *In the methods, you write "To measure the transmission spectrum function in the frequency domain (as shown in Fig. S2(b)), we use 10 transducers to form a rectangle source in the spatial domain, which ensures that the spatial spectrum of the incident field is sufficiently wide but don't overlap with the nonzero diffraction order." I don't see such detailed information about the source in Fig. S2(b), where there is only a point source. And in any case, it seems like there should be more detail on this point.*

b. *Again, since these details are saved for the SM, the figures and the explanations are of lower quality, which I think is very unfortunate for the reader.*

OUR RESPONSE: We appreciate the Referee's suggestion and have modified Fig. S2(b) by

showing the transducer array. Also, we have modified the Methods with a more detailed description of the measurement process.

To measure the transmission spectrum function, an array consisting of ten transducers, with the distance of about 1 cm from each other, is utilized to generate an incident field with sufficiently wide spatial spectrum range which does not overlap with higher diffraction orders, Fig. S2(b). Furthermore, a series of pulses is generated with the frequency spectrum ranging from 7 kHz to 9 kHz.

Furthermore, we use a pair of microphones, with one designated as the reference and the other as the probe. The reference microphone is rigidly positioned at a fixed location within the acoustic field, while the probe microphone scans along the x -direction. At each probe position, the reference and probe microphones record time signals concurrently. By comparing the time signal detected by the probe with the reference signal, the precise time reaching the probe at each measurement point is measured by aligning all the reference signals. In doing so, we first measure the incident acoustic wave $P_{in}(x, t)$ at the scanning line (Fig. S2) without a meta-grating. Then the transmitted acoustic wave $P_{out}(x, t)$ is measured in the presence of the meta-grating. By applying the space-time Fourier transform to the incident (transmitted) acoustic field, we obtain the complex amplitudes of the incident (transmitted) plane waves. This provides the data for the transmission spectral function.

REFEREE 4: *The above also applies to the experimental setup to measure the STVP, albeit less so since there is a sketch in Fig. 3a, and Fig. S6 is of higher quality.*

OUR RESPONSE: We appreciate the Referee's suggestion and have modified the Methods with more detailed description of the STVP measurements.

We use a curved transducer array to simultaneously generate a series of pulses with a Gaussian envelope at the central frequency ω_0 , as shown in Fig. S6(a). The spatial shape of the curved transducer array is described by the function $z = \exp(x^2/\sigma^2) - 1$, whereas the x -intervals between the transducers are 1cm. We put the meta-grating at the distance of 50cm from the transducer array. The pair-microphones method, used to measure the transmission spectrum function, is also applied to measure the STVPs. First, the incident wave $P_{in}(x, t) = S_{in}(x, t)e^{-i\omega_0 t}$ is measured, with the envelope distribution $S_{in}(x, t)$ shown in Figs. S6(b-d). Then, the transmitted wave $P_{out}(x, t) = S_{out}(x, t)e^{-i\omega_0 t}$ is measured by scanning the field at different z -positions.

REFEREE 4: *General comment: Very little of the work focuses on the fact that these are acoustic waves (longitudinal pressure waves) and not transverse electromagnetic waves. This is present in Figure 4, but it is not clear what implication this had on the design of the meta-grating and the simulations necessary to propagate the waves, etc. It would have been nice for there to be more detail on this aspect in the early part of the main article, to set the stage for the work and to provide more context and detail. Are you relying too much on Ref. [33] for that?*

OUR RESPONSE: In fact, our work only uses general properties of *scalar* waves and their scattering, and does not involve any polarization vector properties. In particular, an entirely similar method for the generation STVPs using a meta-grating with broken mirror symmetry has been

described theoretically for optical setup [32]. We do not rely much on Ref. [33] but only use the general fact that STVPs can be constructed similarly for any linear scalar waves. The only difference can be due to different dispersion relations, but in the case of sound, there is the same linear dispersion $\omega = vk$ as for the free-space light.

We have emphasized this scalar-wave universality of our approach in the Introduction of the revised manuscript.

REVIEWERS' COMMENTS

Reviewer #1 (Remarks to the Author):

The authors have addressed my concerns in the revised manuscript and rebuttal letter. I am happy to recommend publication with the minor revisions below.

- "A z -propagating Gaussian pulse impinges on the structure (meta-grating) lying in the $z = 0$ plane and homogeneous along the y -axis. If the meta-grating is mirror-symmetric about the $x = 0$ plane, the phase distribution of the transmitted pulse must also be symmetric about this plane, and thus can bear no phase singularity (vortex) in the (x, z) plane." - This is incorrect. See the phase vortices surrounding the Airy ring singularities as described in Ignatowsky ("Diffraction by a Lens of Arbitrary Aperture", *Trans. Opt. Inst. Petr.*, 1-36, 1919) and Richards and Wolf ("Electromagnetic diffraction in optical systems, ii. structure of the image field in an aplanatic system.", *Proc. Royal Soc. London. Ser. A. Math. Phys. Sci.* 253, 358–379, 1959), which arise from a symmetric aperture about the $x=0$ plane. You do not need mirror symmetry breaking for phase vortex generation in the xz plane. A mirror symmetric aperture just produces vortices that are mirror images of each other - in fact, you observe these vortices in the symmetric structure of Fig 1c. Please revise the necessary condition in the main text and in supplementary S1 to specify what actually enables such vortices or if symmetry-breaking is even needed.

- Please include Figures R1 and R2 somewhere in the main text or supplementary - they are useful.

Reviewer #2 (Remarks to the Author):

The authors addressed my concerns and I can recommend the manuscript for publication.

Reviewer #3 (Remarks to the Author):

The authors had a good revisions and the revised version can be published in my opinion with optional suggestions:

1. Figure 3a is still confusing, it looks like plasmonic waves propagating on a surface... actually, they are pulses in free space. For the schematic it should be more suitable to plot the color map on a sphere (Gaussian pulse) and a torus (STOV) to highlight they are in free space.
2. The simulated results of perturbed pulses can be moved to the main text.

Reviewer #4 (Remarks to the Author):

The authors have satisfactorily responded to my comments and updated the paper (and supplemental material

We thank the Referees for careful reading of our manuscript and relevant remarks. We have addressed these remarks in the revised version of the manuscript and below we respond point by point to the Referees' comments.

REFeree #1: "A z -propagating Gaussian pulse impinges on the structure (meta-grating) lying in the $z = 0$ plane and homogeneous along the y -axis. If the meta-grating is mirror-symmetric about the $x = 0$ plane, the phase distribution of the transmitted pulse must also be symmetric about this plane, and thus can bear no phase singularity (vortex) in the (x, z) plane." - This is incorrect. See the phase vortices surrounding the Airy ring singularities as described in Ignatowsky ("Diffraction by a Lens of Arbitrary Aperture", *Trans. Opt. Inst. Petr.*, 1-36, 1919) and Richards and Wolf ("Electromagnetic diffraction in optical systems, ii. structure of the image field in an aplanatic system.", *Proc. Royal Soc. London. Ser. A. Math. Phys. Sci.* 253, 358–379, 1959), which arise from a symmetric aperture about the $x=0$ plane. You do not need mirror symmetry breaking for phase vortex generation in the xz plane. A mirror symmetric aperture just produces vortices that are mirror images of each other - in fact, you observe these vortices in the symmetric structure of Fig 1c. Please revise the necessary condition in the main text and in supplementary S1 to specify what actually enables such vortices or if symmetry-breaking is even needed.

OUR RESPONSE: We thank the Referee for this comment, and we agree with this point. Indeed, the transmitted field can contain mirror-symmetric vortex-antivortex pairs in the (x, z) plane for the mirror-symmetric meta-grating. However, the goal of our work is to generate an isolated vortex propagating along the z -axis, and the mirror-symmetry breaking is needed for this. We have modified the above sentences and Supplementary Note S1 as suggested by the Referee, and now indicate that we consider a mirror-asymmetric spatiotemporal vortex pulse on the z -axis, which can appear only for mirror-asymmetric meta-grating.

REFeree #1: Please include Figures R1 and R2 somewhere in the main text or supplementary - they are useful.

OUR RESPONSE: We have added these figures to the Supplementary Materials as Fig. S2 and S5.

REFeree #3: Figure 3a is still confusing, it looks like plasmonic waves propagating on a surface... actually, they are pulses in free space. For the schematic it should be more suitable to plot the color map on a sphere (Gaussian pulse) and a torus (STOV) to highlight they are in free space.

OUR RESPONSE: In our experiment, we generate a quasi-2D STVP in a planar waveguide between two glass plates (see Methods). Therefore, the 3D torus configuration is irrelevant. We modify Fig. 3a in this revision to show this point more clearly.

REFeree #3: The simulated results of perturbed pulses can be moved to the main text.

OUR RESPONSE: We prefer to keep it in the Supplementary Materials, because it does not contain any new information compared to the experimental results shown in the main text.